# Structural and Morphological Studies of Pt in the As-Grown and Encapsulated States and Dependency on Film Thickness

**DOI:** 10.3390/nano14080725

**Published:** 2024-04-20

**Authors:** Berkin Nergis, Sondes Bauer, Xiaowei Jin, Lukas Horak, Reinhard Schneider, Vaclav Holy, Klaus Seemann, Sven Ulrich, Tilo Baumbach

**Affiliations:** 1Institute for Photon Science and Synchrotron Radiation, Karlsruhe Institute of Technology, Hermann-von-Helmholtz-Platz 1, 76344 Eggenstein-Leopoldshafen, Germany; berkin.nergis@kit.edu (B.N.); tilo.baumbach@kit.edu (T.B.); 2Laboratory for Electron Microscopy, Karlsruhe Institute of Technology, Engesserstr. 7, 76131 Karlsruhe, Germany; carriorjin@gmail.com (X.J.); reinhard.schneider@kit.edu (R.S.); 3Department of Condensed Matter Physics, Charles University, Ke Karlovu 5, 121 16 Prague, Czech Republic; lukas.horak@matfyz.cuni.cz (L.H.); vaclav.holy@matfyz.cuni.cz (V.H.); 4Institute for Applied Materials, Karlsruhe Institute of Technology, Hermann-von-Helmholtz-Platz 1, 76344 Eggenstein-Leopoldshafen, Germany; klaus.seemann@kit.edu (K.S.); sven.ulrich@kit.edu (S.U.); 5Laboratory for Applications of Synchrotron Radiation, Karlsruhe Institute of Technology, Kaiserstr. 12, 76131 Karlsruhe, Germany

**Keywords:** platinum, bottom electrode, PLD, hillock formation, dewetting, encapsulated platinum, X-ray diffraction, atomic force microscopy, electron microscopy

## Abstract

The morphology and crystal structure of Pt films grown by pulsed laser deposition (PLD) on yttria-stabilized zirconia (YSZ)at high temperatures *Tg* = 900 °C was studied for four different film thicknesses varying between 10 and 70 nm. During the subsequent growth of the capping layer, the thermal stability of the Pt was strongly influenced by the Pt film’s thickness. Furthermore, these later affected the film morphology, the crystal structure and hillocks size, and distribution during subsequent growth at *Tg* = 900 °C for a long duration. The modifications in the morphology as well as in the structure of the Pt film without a capping layer, named also as the as-grown and encapsulated layers in the bilayer system, were examined by a combination of microscopic and scattering methods. The increase in the thickness of the deposited Pt film brought three competitive phenomena into occurrence, such as 3D–2D morphological transition, dewetting, and hillock formation. The degree of coverage, film continuity, and the crystal quality of the Pt film were significantly improved by increasing the deposition time. An optimum Pt film thickness of 70 nm was found to be suitable for obtaining a hillock-free Pt bottom electrode which also withstood the dewetting phenomena revealed during the subsequent growth of capping layers. This achievement is crucial for the deposition of functional bottom electrodes in ferroelectric and multiferroic heterostructure systems.

## 1. Introduction

The optimization process of multiferroic heterostructures of our interest, based on bilayer systems of ferromagnetic materials such as the hexaferrite BaFe_12_O_19_ and ferroelectric materials such as LuFeO_3_ and YbFeO_3_, crucially requires first a high-quality bottom electrode, which remains stable at elevated growth temperatures *Tg* > 900 °C in an oxidizing atmosphere during the pulsed laser deposition (PLD) growth of the above-mentioned subsequent layers [1,2,3,4,5]. Platinum is one of the noble metals, and is widely used in the application field of metal-ferroelectric-metals (MFMs) [6,7,8], ferroelectric memories [9], and microelectromechanical systems (MEMSs) [10] due to its high melting point, inertness in hard atmospheres, and low resistivity. Several studies have reported the effects of the structure and morphology of the bottom electrode on the performance of ferroelectric films and MEMSs [7,8,10]. Roshchin et al. [7] have demonstrated that imperfections like pores, cavities, and interdiffusion, which can occur at the ferroelectric bottom electrode, significantly influence the electrical properties of MFM systems. Other studies have dealt with the disturbance of the thermal stability of Pt by hillock formation and dewetting (i.e., agglomeration in the literature) after heat treatment [11,12,13,14,15,16,17,18]. The appearance of hillocks was mostly studied during heat treatment of Pt, where the growth temperature *Tg* was varied between room temperature (RT) and 500 °C. There, it has been found that hillock formation is mainly related to annealing, and the size and density of hillocks were dependent on *Tg* which affects the amount and nature of the internal stress (compressive or tensile) generated in the Pt film during growth [14,15].

Pt hillocks are considered as a major issue as they can interconnect the top and bottom electrodes and consequently lead to short circuits in capacitors, as demonstrated by Kweon et al. [15]. These morphological changes were also recorded during the annealing of other metals such as Al, Ag, Cu, and Au [19,20,21,22,23]. Furthermore, former research pointed out that the agglomeration process, which includes nucleation and hole growth steps, follows either capillary or fractal agglomeration processes depending on the type of metal [21].

Moreover, to the best of our knowledge, very few studies were concerned about the double dependency of hillock formation and dewetting phenomena on film thickness and growth temperature *Tg* during the growth and annealing of metallic layers such as Al [19] and Pt [12,24] on insulator substrates, where Pt was grown at *Tg* = RT. Nevertheless, Hren et al. [13] mentioned that the mechanisms of hillock formation for Al and Pt are quite different. In the case of the metal Al, the formation mechanism seems to be based on three steps: the extrusion of the entire grain from the film surface, followed by a regrowth of new grains at triple-junction points, and finalized by the addition of new material to existing grains on the top of the surface. While in the case of Pt, the mechanism of hillock formation was rather a regrowth and consolidation process of Pt grains, leading to the appearance of bumps on the film surface [13]. Furthermore, more effort was spent on the understanding of the hillock growth mechanism in Al films. In fact, by means of transmission electron microscopy cross-sections, hillock growth in Al films was demonstrated to be controlled via diffusion along grain boundaries which involved a mass transport along grain boundaries, along the surface of the grown film, as well as along the interface between the film and the substrate [25].

Even though hillock formation and agglomeration are Pt film instability issues that degrade the material properties, their mechanisms are different. While hillocks are the result of a delocalized strain field in the thin film, agglomeration is caused by local defects which induce hole formation and metal film rupture [16].

In order to improve the adhesion of Pt to oxide substrates, Ciftyürek et al. employed a variety of adhesion layers like Ta, Ti, and Zr as well as multilayers (for example Hf + Zr) as coatings for the Pt bottom electrode. Their aim was to protect the conducting electrode of MEMS devices against the dewetting of the film during the operation at high temperatures (>1000 °C) [17]. Pt bottom electrodes were deposited on different types of insulator substrates such as Al_2_O_3_ [2,3,17], SiO_2_ [11,12,13,14,15], and YSZ [16,18,24] with various deposition methods such as sputtering [13,14,15,17,18,26,27], ion beam evaporation [10,12,16], PLD [3,28], and others [11].

The difference in the lattice mismatch and the thermal expansion coefficient between platinum and the different studied substrates affects the growth mechanism and Pt film morphology, as well as hillock formation and agglomeration phenomena.

Moreover, one of the few attempts [15] which dealt with the morphological changes in Pt electrodes incorporated between the substrate and the overgrown layer was about the effect of the SrBi_2_Ta_2_O_9_ capping layer on Pt hillock formation by means of an SEM cross-section.

However, several studies [10,11,12,17,24,29] were concerned about morphological changes during the annealing of the Pt bottom electrode without a capping layer, termed in our investigation as the subsequently grown layer. To the best of our knowledge, there have been no detailed structural, morphological, or chemical analyses about the changes happening in the bottom layer in the as-grown state and after being overgrown with a capping layer.

In this work, we investigate the morphology and the microstructure of Pt films which have different thicknesses in two states: the as-grown state and the encapsulated state, respectively, in bilayer systems using combined and complementary methods such as atomic force microscopy (AFM), scanning electron microscopy (SEM) in combination with energy-dispersive X-ray spectroscopy (EDXS), transmission electron microscopy (TEM), and high-resolution X-ray diffraction (HRXRD).

The present study deals with the description of the modification in the morphology and the structure of the Pt bottom electrode as being capped with the subsequent grown layer in a systematic comparison with the Pt uncapped layer using the same methodology. This will be discussed in detail in the next sections.

## 2. Materials and Methods

### 2.1. PLD Growth of Samples

The Pt films were deposited on yttria-stabilized zirconia (YSZ) with an (111) orientation in a PLD chamber in a vacuum environment after heating the substrate to *Tg* = 900 °C using a different number of shots, namely 1647, 4100, 8235, 12350, and 16470, with a laser frequency of 5 Hz and an energy pulse of 60 mJ (i.e., fluence F = 16 J/cm^2^). Prior to the growth, the different substrates were cleaned with isopropanol and then annealed in the furnace for 2 h at a temperature of 1200 °C in order to have low roughness and a good terrace morphology. The target was separated by 35 mm from the substrate. The Pt films in the as-grown state with thicknesses of 10 nm to 70 nm are named in the manuscript as follows: Pt_Th10 nm, Pt_Th25 nm, Pt_Th40 nm, Pt_Th55 nm, and Pt_Th70 nm. The samples were cooled slowly at 5 °C/min from *Tg* = 900 °C to room temperature (RT). The substrates used for the Pt growth were 8 × 8 mm^2^ in size. We could divide the grown Pt films into two pieces for the subsequent treatment leading to the two different states: namely, as-grown and encapsulated. In the case of one of the pieces with a size of 4 mm × 8 mm, ferroelectric (FE) YbFeO_3_ layers or ferromagnetic (FM) layers BaFe_12_O_19_ were grown on the different Pt bottom electrodes mentioned above. For this purpose, the template (Pt/YSZ) was heated from RT to *Tg* = 900 °C with a heating rate of 25 °C/min. The growth of the subsequent YbFeO_3_ layers was carried out in an oxygen atmosphere at a pressure of 400 mTorr and with a laser frequency of 1 Hz. The deposition time was about 5 h. The Pt layer became encapsulated in the multilayer system between the YSZ and the FE or FM layers with thicknesses of about 100 nm, which are termed as the capping layers (CLs). In the manuscript, the corresponding samples with encapsulated Pt films of different thicknesses are named EncPt_Th10 nm, EncPt_Th25 nm, EncPt_Th40 nm, EncPt_Th55 nm, and EncPt_Th70 nm.

### 2.2. Atomic Force Microscopy (AFM)

Ex-situ AFM topography measurements of the nine Pt layers were carried out in tapping mode with a Bruker Dimension ICON (Bruker, Karlsruhe, Germany). As sensors, we used OPUS AC160-NA cantilevers (NanoAndMore, Wetzlar, Germany), with force constants of 26 N/m and resonance frequencies of 300 kHz. The analysis of the AFM images was performed using the NanoScope software package v2.0 (Bruker, Karlsruhe, Germany). All the derived topographical parameters are summarized in Table 1.

### 2.3. Scanning Electron Microscopy (SEM) and Transmission Electron Microscopy (TEM)

The surface topography and the chemical composition of these nine Pt layers were characterized by SEM imaging combined with energy-dispersive X-ray spectroscopy (EDXS) using an FEI Dual beam Helios G4 FX microscope (Thermofisher Scientific, Waltham, MA, USA). For secondary electron (SE) imaging of the sample surface with 0 and 50 degrees of inclination, in the following called SE_0° and SE_50°, respectively, an Everhart-Thornley detector (ETD) (Massachusetts, Bruker, USA) was used. The microscope was operated at a 10 kV accelerating voltage in the so-called field-free mode with a beam current of approximately 25 pA. In addition, to obtain element-specific information, backscattered electron (BSE) images were taken by means of a semiconductor (pn-diode) detector. Moreover, chemical analyses were performed by EDXS at a primary electron energy of 20 keV and a beam current of 0.4 nA, applying a Bruker (Billerica, MA, USA) system of the type QUANTAX 400 with a silicon-drift detector (SDD) XFlash 6 (Bruker, Billerica, MA, USA). In particular, for the samples without inclination (EDX_0°) the two-dimensional Pt distribution was imaged via mapping, where the acquisition time per map was about 5 min. By employing the ESPRIT 2.3 software, the raw-data X-ray maps were quantified using the thin-film approximation of Cliff–Lorimer [30] to obtain element-concentration maps.

For TEM inspection of the different samples (Pt_Th25 nm Pt_Th40 nm, Pt_Th55 nm, and Pt_Th70 nm in the as-grown state and the films EncPt_Th25 nm EncPt_Th40 nm, EncPt_Th55 nm, and EncPt_Th70 nm in the encapsulated state), cross-sectional specimens were prepared by focused ion beam (FIB) milling using the FEI Dual beam Helios G4 FX microscope. Prior to FIB preparation, a thin gold layer was sputtered on the sample surface in order to reduce ion-beam damage of the heterostructures. Subsequently, standard FIB preparation of TEM lamellae was performed, where a Pt protection layer was deposited on top of the samples. Then, coarse FIB milling was carried out at a primary ion energy of 5 keV. The lamellae were attached to Cu lift-out grids and finally polished by a Ga+-ion beam with a low energy rate of 1 keV to minimize Ga+ implantation and material amorphization.

TEM investigations of all the above-mentioned samples were carried out on an aberration-corrected FEI Titan 80–300 microscope (Thermofisher Scientific, Waltham, MA, USA). This TEM has a thermally assisted field emission cathode (Schottky emitter) and was operated at 300 kV high voltage. For image recording, the microscope was equipped with a 4 k × 4 k CMOS camera of the F436 type (TVIPS). TEM bright-field imaging was performed to obtain information about general layer properties, e.g., layer thickness and crystal structure.

### 2.4. Degree of Coverage

The values of the degree of coverage were obtained by processing BSE_0° images (see Appendix A). In detail, the Weka Trainable Segmentation plug-in of the ImageJ (v1.54) software [31] was used to classify Pt and substrate (YSZ) regions. Dark contrast groves were marked as the substrate and regions with bright contrast as Pt. The software was iteratively trained by comparing the marked regions with the original image. The Weka Trainable Segmentation plug-in allows for the identifying of the variable contrast regions. Image defects (such as contrast profile, charging contrast, etc.) could be eliminated by the iterative training opportunity of the plug-in. The segmentation results were converted into binary images and the area fractions were measured by ImageJ and determined as the degree of coverage.

### 2.5. Simulation of X-ray Reflectivity (XRR) Curves

X-ray reflectivity (XRR) measurements were performed by means of an X-ray diffractometer Rigaku using a rotating Cu anode for a monochromatic beam with a wavelength of 1.54056 Å. The calculation of the theoretical XRR curves is based on the dynamical theory of X-ray scattering. This theory assumes a system of perfectly flat, homogeneous slabs with a given thickness and density, grown on a perfectly flat substrate. The roughness of the surfaces and interfaces are typically incorporated using the distorted-wave Born approximation approach. However, this approximation is only valid when the perturbation to the system is small, meaning that the root-mean-squared roughness is much smaller than the layer thickness. As evidenced by TEM images, in our case, the perturbation is considerably more severe, preventing the use of a simple model involving just one rough Pt epilayer or even two layers, including the encapsulating layer. Nevertheless, if inhomogeneities such as voids, pores, and thickness fluctuations are much smaller than the coherent length, the original calculation approach based on the dynamical theory can still be applied. In this scenario, the sample needs to be described as a multilayer consisting of numerous very thin, homogeneous slabs, each with an averaged electron density within a given depth. In essence, the depth profile of the mean electron density provides a complete description of the sample.

Consequently, this model involves a relatively high number of parameters (mean density at specified depths), which complicates the convergence of the optimization process. The experimental data were fitted using a trust-region-reflective optimization algorithm. The resulting depth profiles should be considered representative, as slightly different solutions (differing in fine features such as steps or weak oscillations) are obtained for each separate optimization run. Nonetheless, all these solutions exhibit the same characteristics, revealing the mean layer thickness, transition regions at interfaces, mean density, and depth density variation.

### 2.6. High-Resolution X-ray Diffraction Reciprocal Space Mapping (HR-RSM)

High-resolution X-ray diffraction reciprocal space mapping (HR-RSM) was performed at the NANO beamline at the synchrotron facility at KARA, Karlsruhe, Germany, using a well-collimated and monochromatic beam with an energy rate of 15 keV and a wavelength λ of 0.826 Å. The symmetric reflections Pt111, Pt222, and Pt333 and asymmetric reflections Pt224 and Pt331 were measured by rocking the sample around the Bragg angle using a linear Mythen detector positioned at Bragg angles corresponding to the different reflections. From the different HR-XRD maps, we derived the angular as well as the radial diffraction profiles to determine the misorientation and the mean values of the vertical strain distribution <ε_⟂_>, respectively. The variation of these latter values with the Pt film thickness *Th_Pt_* will be discussed later. The HR-RSMs of the Pt111 and Pt222 reflections of the as-grown and encapsulated states are given in Appendix A for Pt_Th10 nm, Pt_Th25 nm, PtTh40 nm, Pt_Th55 nm, and Pt_Th70 nm. In addition, the radial and the angular cuts were also compared in Appendix A.

Determination of the interplanar d-spacing of Pt grown on YSZ(111)

From the different HR-RSM data of the (111), (222), and (333) reflections of platinum given in Appendix A, we derived the coordinates of the diffraction vectors corresponding to the maximum intensity of the diffraction peaks *Q_Z_*. Here, we assume that Pt has a nearly cubic crystal structure with a small rhombohedral distortion, such that the lattice angle is *α* = 90 + *δ*, where δ is the distortion angle. In this case, the <*d*_111_> spacing was determined as the mean value of *d*_111_, which was derived from the measured *Qz*, the coordinates of the (111), (222), and (333) reflections, while the d_11-2_ spacing was derived from the *Qx*_331_ coordinates corresponding to the (331) reflection (see Appendix A) in accordance with the following:(1)$${<d}_{hkl}>=\frac{2\pi }{{l*Q}_{z}}$$
where (hkl) = (111), (222), and (333),
(2)$$<d_{11-2}>=\frac{3}{2}\frac{2\pi }{Q_{x331}}$$
(3)$${\Delta d}_{hkl}=d_{hkl\_cal}-\;d_{hkl\_meas}$$
where *d_hkl_cal_* is the d-spacing calculated for the rhombohedral structure and *d_hkl_meas_* is the d-spacing derived from the *Q* coordinates using Equations (1) and (2). By minimizing the Δ*d_hkl_*, it was possible to determine the distortion angle *δ* and the lattice parameter *a* of the rhombohedral structure.

Determination of the out-of-plane residual strain

As a result, the in-plane and the out-of-plane directions of the Pt unit cell grown on YSZ with a (111) orientation correspond to the (111) and (11-2) directions, respectively. In this case, the out-of-plane residual strain ε_⊥_ will be determined as the following:(4)$$\varepsilon_{⊥}=\frac{{d_{111}^{Pt}-d}_{111}^{Bulk}}{d_{111}^{bulk}}$$
where $d_{11-2}^{Bulk}$ and $d_{111}^{Bulk}$ correspond to the d-spacing *d*_11-2_ and *d*_111_ of the Pt in the bulk.

### 2.7. Electrical Conductivity

The electrical resistance of the films was measured using a conventional multimeter (2-point resistance measurement). To approximate the 2-point resistance measurement into the 4-point sheet resistance, 7 measurements were conducted on each 4 mm × 8 mm thin film sample at varying probe positions (i.e., 2 diagonal measurements, 2 measurements along the long edge, and 3 measurements along the short edge). The measured resistance was utilized to calculate the conductivity. The conductivity calculation of the films was performed using the Finite Element Method Magnetics (FEMM 4.2) software [32]. All the values of probe positions, probe conductivity, film geometry, film thickness, and film conductivity were given as input values to the FEMM 4.2 software, and resistances were taken as output values. The resistance output was approached to the measurement value by iterating the initial given conductivity. The given conductivity value, at which the output and the measured resistance values have a difference of less than ±1% to each other, was determined as the conductivity. Seven measurement values were then averaged, and the average value was assigned as the sheet conductivity. The conductivity values were only determined for the Pt films Pt_Th10 nm, Pt_Th25 nm, Pt_Th40 nm, Pt_Th55 nm, and Pt_Th70 nm in the as-grown state and are summarized in Table 1, row #27.

## 3. Results and Discussion

The understanding of the capping layer effect on the morphological and structural changes in the Pt bottom electrode crucially requires a comparison with Pt as an uncapped layer using the same methodology. Even though the study of Pt without a capping layer was intensively studied [10,11,12,16,24,29], the encapsulated Pt layer was almost not investigated. Therefore, our study was carried out in a systematic way on Pt films with different thicknesses in uncapped (i.e., as-grown) and in the encapsulated states.

### 3.1. The Dependency of the Morphology and Degree of Coverage on the Deposition Time in the As-Grown State

Figure 1(b1–f1) shows the AFM micrographs corresponding to Pt films grown at *Tg* = 900 °C with different thicknesses, *Th_Pt_* = 10, 25, 40, 55, and 70 nm, respectively. In comparison, Figure 1(a1) confirms the terrace morphology of YSZ(111) prior to the growth after cleaning and thermal treatment. Pt_Th10 nm displays 3D island morphology with a mean particle size of 47 nm (see Figure 1(b1) and Table 1, line 11, column#1). The increase in the deposition time for Pt has brought several mechanisms like nucleation, coalescence, and elongation transition into competition. By increasing the adatom arrival rate, the formation of islands took place. Due to the high *Tg*, the diffusivity of the adspecies on the substrate surface is considered high; this in turn favors the adatom incorporation into existing islands at the expense of nucleating new ones. The further increase in the deposition time contributes to the enlargement of the islands of the critical size for the occurance of the coalescence and the formation of elongated chains for the case of Pt_Th25 nm (see Figure 1(c1)). The width of the channels varies between 41 nm and 141 nm (Table 1, line 12, column#2), while the mean size of the Pt-free holes is about 41 nm (Table 1, line 17, column#2). The increase in *Th_Pt_* to 40 nm enhances coalescence, which leads to an enlargement of the channels and at the same time to a reduction in the separating spaces as well as to a shortening of the length of the valleys (see Figure 1(d1)). Figure 1(e1) shows the morphology of the sample Pt_Th55 nm as a network containing randomly distributed holes, where the size of these holes varies between 18 nm and 111 nm (Table 1, line 17, column#4). The increase in *Th_Pt_* from 55 to 70 nm illustrates a clear modification from a network with holes to a continuous film in the case of Pt_Th70 nm with a roughness of *R_a_* = 0.6 nm, containing still small holes with a mean size of about 46 nm (see Figure 1(f1) and Table 1, lines 10, 18, column#5).

Besides the changes in the Pt film morphology, the PLD growth of Pt on the YSZ substrate was accompanied by the formation of hillocks. These later appeared as bumps in the AFM micrographs with different contrasts. In order to demonstrate the height variation of the hillocks, three profiles were derived from red, black, and green dashed lines as drawn in the AFM micrographs (see Figure 1(b1–f1)). The different profiles are given in Figure 1(b2–f2) for the samples Pt_Th10 nm, Pt_Th25 nm, Pt_Th40 nm, Pt_Th55 nm, and Pt_Th70 nm, respectively, where it is possible to compare the height variation of the Pt films due to the hillocks (i.e., *I_max_*) and due to the voids (i.e., *I_min_*). As an example, in the case of sample Pt_Th10 nm, the hillock height varies between 5.3 and 44.7 nm (see Table 1, line 24, column#1).

It is worthwhile to mention that the morphology of a Pt film grown on a YSZ substrate including the formation of hillocks and voids depends on the film thickness *Th_Pt_* as well as on the growth temperature *Tg*. For example, a percolating network similar to the morphology of Pt_Th25 nm grown by PLD at *Tg* = 900 °C was reported by Ryll et al. [24] by annealing the sputtered Pt initially grown at a low temperature. Furthermore, the morphology of a Pt film with *Th_Pt_* = 50 nm sputtered at a low temperature and annealed between 625 and 800 °C did not reveal any hillock formation but a dewetting issue. Contrarily, the study of Beck et al. [28] about the microstructure of a Pt film with *Th_Pt_* = 400 nm grown on YSZ by PLD at *Tg* = 400 °C revealed the formation of triangular and hexagonal hillocks. This comparison emphasizes that the deposition methods and the growth conditions strongly influence the morphology and the microstructure of Pt films grown on a YSZ substrate. On the other hand, preserving the same deposition method and deposition parameters disclosed a reproducible morphology similar to the one shown by AFM micrographs in Figure 1. However, changing the growth temperature (i.e., *T_g_* = 300 °C to *T_g_* = 900 °C) or using a different substrate (i.e., Al_2_O_3_) would modify the morphology and the structure, as clarified in our previous work [3].

Figure 2a–e show SE_0° images of the 1.2 × 1 µm^2^ areas with a zero-degree angle (i.e., the incoming electron beam is perpendicular to the sample surface) for the same specimens as investigated by AFM (see Figure 1(b1–f1)). Similarly, to AFM, SE_0° images confirm a transition in morphology from 3D islands to 2D layers as the *Th_Pt_* increased from 10 to 25 nm and, therefore, demonstrate an enhancement of coalescence with the deposition time (*Th_Pt_* > 25 nm). For the sample PtTh70 nm, the SE_0° image also shows the formation of a smooth and continuous Pt film. Figure 2f–j represent corresponding Pt elemental EDX_0° maps where the dark regions indicate Pt-free regions in the shape of channels, like for Pt_Th25 nm, and of holes for Pt_Th40 nm and Pt_Th55 nm. However, from SE_0° images and their corresponding EDX_0° maps, it is quite difficult to identify the location of the hillocks and to measure their sizes. To promote the visibility of the hillocks, the samples were inclined at different angles; Figure 2k–o show SE_50° images of 5 µm × 5 µm in size, which were obtained with an inclination angle of 50 degrees for the best visibility of hillocks, where the hillocks were encountered with red circles. These later were used for the quantification of the area density of hillocks. In the SE_50° images, the hillocks appear as bumps located on the surface. It has to be mentioned that in the SE_50° images, the height visibility of hillocks is improved in comparison with SE_0° images, but it is still not as pronounced as in the AFM images. In order to enhance the visibility of the hillocks, causing a change in the surface topography, we utilized the backscattered electron imaging method without a sample tilt (i.e., BSE_0°). This technique enables the collection of electrons where the local electron density depends on the inclination angle of the scattering object [33] relative to the incident electron beam. The hillocks, with their pyramidal-like shape and facets being differently inclined with respect to the incident electron beam and having a different packing density to the crystal lattice, scatter the electrons more or less back to the BSE detector, consequently generating bright spots in the BSE_0° images, as can be seen in Figure 2p–t. For sample Pt_Th40 nm, the comparison of Figure 2c,m,r, which correspond to SE_0°, SE_50°, and BSE_0° images, respectively, demonstrates that the highest contrast of hillocks was obtained in the BSE_0° images. This applies also to the case of the Pt_Th55 nm sample. As a conclusion, BSE_0° images offer the possibility to identify hillocks and to follow their evolution as a function of Pt film thickness *Th_Pt_*.

The hillock densities were determined from SE_50° images as well as from AFM and BSE_0° images, and they are compared in Table 1, rows #22 and 23. Furthermore, BSE_0° images were analyzed to derive the degree of coverage corresponding to Pt in the as-grown state as indicated in a percentage in Appendix A. Obviously, the degree of coverage significantly increased from 66.2% in the case of Pt_Th10 nm to 99.8% for Pt-Th70 nm. This consequently had a strong impact on the measured conductivity values which developed from 0 to 3.69 × 106 (1/Ohm) (see Table 1, row #27). This result is consistent with the finding of Bauer et al. [3] regarding the improvement of the electrical conductivity of Pt grown on sapphire with the increase in the deposition rate.

In contrast to Bauer et al.’s previous study [3] carried out on a Pt_Th40 nm film grown on Al_2_O_3_, the obtained morphology from Pt_Th40 nm grown on YSZ at *Tg* = 900 °C was different and unavoidably exhibited hillock growth even for higher thicknesses such as Pt_Th55 nm (see Figure 2r,s). The reported absence of the hillocks in the case the Pt_Th40 nm film grown on Al_2_O_3_ could be due to the difference in coefficients of thermal expansion (CTEs) and lattice misfits. Furthermore, the hillock formation was recognized to be dependent on the compressive stress during heating and on the stress relaxation film during cooling [14]. Carneiro et al. [34] investigated as to whether residual stress is dependent on the CTE of the film *α_f_* being higher (respectively lower) than the CTE of the substrate *α_S_*. Their analysis confirmed that a thin film experienced tensile stress if $\alpha_{f}>\alpha_{s}$ and compressive stress for the case of $\alpha_{f}<\alpha_{s}$. As the CTE of Pt ($\alpha_{Pt}$ = 8.9 × 10^−6^/°C) [35] was smaller than the CTE of YSZ $\alpha_{YSZ}$ = 10.5 × 10^−6^/°C) [36], the film was compressively stressed, in contrast to a previous study of Bauer et al. [3] where $\alpha_{Al2O3}$~7 × 10^−6^/°C [37].

Figure 3a shows that the hillock density decreased with the thickness *Th_Pt_* of the deposited Pt film, while simultaneously the hillock volume increased. It appears that for Pt_Th10 nm, the hillocks achieved the highest density and the lowest volume. The behavior of the hillock density was found to be notably influenced by the growth morphology, i.e., the transition from a 3D island to 2D layer growth, when *Th_Pt_* exceeded 10 nm. In the 2D layer growth regime in Figure 3a, Pt_Th25 nm is labeled as “NH” as it did not contain hillocks and the surface morphology was simply composed of channels. Pt_Ths40 nm, which is labeled as “H” (i.e., hillocks), had the highest hillock density [/µm^2^], while the sample Pt_Th55 nm, that is also labeled by “H”, had the largest hillock volume.

The recorded decrease in hillock density is probably related to the reduction in the number of grain boundaries, which was induced by the increase in deposition time, leading to an extension of the lateral size of platinum regions. This correlation will be discussed in more detail in the next chapter.

Figure 3b displays the measured and simulated XRR curves for the samples Pt_Th10 nm, Pt_Th25 nm, Pt_Th40 nm, Pt_Th55 nm, and Pt_Th70 nm. The film thicknesses as well as the roughness Rax were derived from the fitting of the experimental curves (see Table 1, lines 7, 9). In the as-grown state, the oscillation period decreased as the *Th_Pt_* increased, and the derived thicknesses *Th_Pt_* of about 10, 25, 44, 55, and 70 nm from the fitting procedure confirm the expected values from the number of shots mentioned in the growth section above. We found that in the 2D morphology, the surface roughness gradually increased from *Ra^x^* = 1.83 nm for Pt_Th25 nm to *Ra^x^* = 4.24 nm for Pt_Th40 nm and *Ra^x^* = 8.94 nm for Pt_Th55 nm, while it decreased to *Ra^x^* = 0.6 nm for Pt_Th70 nm. This behavior is well correlated with the variation of the hillock volume as a function of *Th_Pt_*, which achieved its maximum value for the sample Pt_Th55 nm (cf. Figure 3a).

### 3.2. The Dependency of the Crystalline Structure on the Deposition Time in the As-Grown State

X-ray diffraction patterns were recorded for different spots of symmetric reflections, namely Pt111, Pt222, and Pt333, and also for the asymmetric reflections Pt224 and Pt331. An overview of the diffraction patterns for the symmetric reflections Pt111 and Pt222 as well as asymmetric reflections Pt331 and YSZ224 is presented in Appendix A, respectively. This covers the diffraction patterns recorded in the as-grown state of the films (see Appendix A). The coordinates of the peak positions which correspond to the maximum intensities Qang and *Q_rad_,* were derived from the patterns of the symmetric and asymmetric reflections. The coordinates were utilized for the determination of the interplanar d-spacings *d*_111_ and *d*_11-2_ along the out-of-plane and the in-plane direction of the rhombohedral structure following the Equations (1) and (2).

The angular and radial diffraction profiles of the Pt111 reflection were derived from the patterns of the different as-grown Pt films and are compared in Appendix A. The comparison of the angular broadening in Appendix A shows a decrease with *Th_Pt_*, indicating either an increase in the lateral coherence length *L_H_* and/or a decrease in the degree of misorientation of the mosaic blocks. Furthermore, the comparison of the radial diffraction profiles in Appendix A shows an oscillation of Kiessig fringes for Pt_Th10 nm and Pt_Th25 nm, which permits us to directly derive the mean value of the vertical size of coherently diffracting crystal blocks *Lv*. Here, we found that *Lv* as determined from the oscillation fringes had the same order of magnitude as *Th_Pt_*. However, the existence of hillocks acted as a disturbance for the coherent diffraction and contributed to the damping of the oscillation fringes in the case of Pt_Th40 nm and Pt_Th55 nm. This was not the case for the sample Pt_Th70 nm, where the radial diffraction profiles exhibited oscillation fringes with a smaller periodicity corresponding to a film thickness of *Th_Pt_* = 70 nm.

Figure 4a,b compare the radial and angular diffraction profiles of Pt333 for the samples Pt_Th10 nm, Pt_Th25 nm, Pt_Th40 nm, Pt4_Th55 nm, and Pt_Th70 nm. Figure 4a also encloses the signal of the YSZ444 reflection of the substrate that is expectedly superposed. This ensures us a reliable determination of the peak positions Qz and therefore of the d_111_ d-spacing. In the inset of Figure 4a, the maximum intensity I_max_ of the Pt333 reflection linearly increases with the deposition time (respectively Th_Pt_) with two different slopes when Th_Pt_ varies from 10 to 70 nm. Furthermore, the slope of the curve gets steeper as Th_Pt_ increases from 55 to 70 nm.

The increase in I_max_ is strictly related to the increase in the Pt film thickness Th_Pt_. This is well correlated with the increase in the degree of coverage as demonstrated in Appendix A. Oppositely, the Qz@I_max_ decreased when Th_Pt_ varied from 10 to 40 nm and remained unchanged.

In the following, to demonstrate these experimental findings, we selected the variation of the diffraction peak Pt333, which was measured together with the YSZ444 reflection of the substrate taken as a reference.

Figure 4b demonstrates the reduction in angular broadening with *Th_Pt_*, which reflects an improvement of the crystal quality of the growing Pt film in terms of decreasing defect density as *Th_Pt_* increases. This is in accordance with the enhancement of the degree of coverage determined in Appendix A for the as-grown state.

Figure 4c depicts the variation of the distortion angle δ of the Pt rhombohedral structure as well as lattice parameter a with *Th_Pt_*. Both structural values follow the same behavior versus *Th_Pt_*. Independent of the hillock density and hillock volume, which varied between Pt_Th10 nm, Pt_Th40 nm, and Pt4_Th55 nm, the degree of the distortion was found to be very comparable and in the range of δ = 0.3 degrees. However, the distortion angle was only affected by the hillock formation, which represents an obstacle for the distortion phenomena. In fact, for the sample Pt_Th25 nm, which did not reveal hillocks (“NH”), the distortion angle was δ = 0.7°, i.e., twice the value determined for the samples Pt_Th10 nm, Pt_Th40 nm, and Pt4_Th55 nm, where hillocks were measurable in the films. Figure 4d shows the evolution of the tensile out-of-plane residual strain ε_⟂_ as well as the interplanar d-spacing d_111_ of the Pt (111) lattice planes as a function of Th_Pt_. Both ε_⟂_ and d_111_ display a similar behavior since they are related by Equation (4).

The in-situ stress measurement of Matsui et al. [14] demonstrated that Pt films sputtered on silicon at higher temperatures than *Tg* = 300 °C and had tensile residual stress due to hillock formation, which released the stress formed during deposition. Due to the discontinuity of the Pt film, which was induced by the formation of voids at lower thickness, we cannot rely on the estimated values of the in-plane residual strain. In the following section, we restrict our analysis to the estimated vertical residual strain *ε*_⟂_ in growth direction. This is worthwhile to remind us about the different phenomena which overlap and compete with the increase in deposition time (i.e., *Th_Pt_*) such as 3D–2D morphology transition, hillock formation, and the generation of voids. It appears that the Pt film sample Pt_Th40 nm had a 2D morphology as demonstrated by AFM in Figure 1(d1) and also enclosed the highest hillock density (see Figure 3). Additionally, Pt_Th40 nm displayed the highest value of residual strain *ε*_⟂_ = 0.0035 and a lattice spacing *d*_111_ = 2.274 Å. In the 2D morphology, by increasing *Th_Pt_* the degree of coverage increased and the hillock density decreased. This led to a shrinking of (111) lattice plane d-spacing and therefore to a reduction in the corresponding <*ε*_⟂_> (Figure 4d).

The morphology in terms of continuity and homogeneity for the case of Pt_Th70 nm was found to release the residual strain in the film and therefore to omit the hillock formation and reduce the expansion of the in-plane lattice parameter and the degree of the distortion (Figure 4c,d).

The fitting of the radial and angular broadening of the different reflection orders Pt111, Pt222, and Pt333 with the Pseudo-Voigt functions [38] (see Appendix A) enabled us to follow the variation of *FWHM_Rad_* and *FWHM_Ang_* with the reflection order for the different Pt film thicknesses as shown in Figure 5a,b. These later demonstrate a decrease in the radial and in the angular broadening which indicates an improvement of the crystal quality with increasing *Th_Pt_*.

The evaluation of the Williamson–Hall [39] (WH) plots of *FWHM_Ang_* (respectively *FWHM_Rad_*) as a function of the reflection order gives the mean value of the lateral size *L_H_* (respectively vertical size *L_V_*) from the intercept and the degree of misorientation *α* (respectively mean value of vertical strain distribution *ε*) from the slope. We found that the increase in the Pt film thickness *Th_Pt_* led to an increase in the *L_H_* and *L_V_* values (see Figure 5c).

The mechanism controlling the hillock formation was discussed for metallic layers such as Al [40], Cr [41], and Pt [13]. The grain boundaries diffusion-based mechanism was proposed as the stress relief which occured during the growth of the Pt layer by the migration of the Pt along the grain boundaries toward the film surface [42]. This intensive migration along the grain boundaries culminated in hillock formation. This enables us to conclude that the presence of a high number of grain boundaries represents nucleation centers for hillock formation. Figure 5c demonstrates an increase in the lateral size *L_H_* of the Pt crystal mosaic blocks which was generated either by the motion or by the disappearance of grain boundaries. This could result from the filling of these diffusions paths during the enhancement of the Pt deposition flux, meaning the increase in Pt film thickness *Th_Pt_*.

Our investigation demonstrated a correlation between the increase in the lateral size which is accommodated by the increase in Pt film thickness and the reduction in hillock density as shown in Figure 3a and Figure 5c.

Furthermore, the compressive differential strain produced by the Pt growth because of the different thermal coefficients of the Pt and the YSZ substrates (i.e., 8.9 × 10^−6^ K^−1^ and 10.5 × 10^−6^ K^−1^ at 300 K, respectively) can be released either by a grain boundaries diffusion-based mechanism of hillocks [41], such as in the Pt_Th40 nm and Pt_Th55 nm samples, or by grain boundaries grooving [24,43], rather than hillock formation which probably occurs in the case of Pt_Th25 nm, leading to the network morphology composed of a high number of distributed channels (see Figure 2b). This consequently can explain the reason for the absence of hillock accompanied by the large number of channels in the case of Pt_Th25 nm. Similar behavior was reported by Ryll et al. [24] in the case of platinum thin film electrodes on a YSZ electrolyte, where the Pt film morphology displayed the same percolating network structure due to the grooving occurring at the grain boundaries.

Simultaneously, the degree of misorientation *α* decreased, which reflects an enhancement of the film quality and a reduction in the defect density (see Figure 5d). However, the mean value of vertical strain distribution <*ε*_⟂_> behaved differently. It reached the maximum for *Th_Pt_* = 40 nm and *Th_Pt_* = 55 nm, corresponding to states labeled by “H” as shown in Figure 5d. As a result, we conclude that the vertical strain distribution became large due to an increase in the hillock height which was estimated to be about 23.6 nm in the case of Pt_Th40 nm and 31.8 nm for Pt_Th55 nm (Table 1, row#24).

### 3.3. Microstructural Changes of the Pt Films during the Subsequent Growth of the Capping Layer

It is worthwhile to emphasize that the investigation of the Pt microstructure was not as easily accessible in the uncapped state due to the overlap of signal coming out of the capping layer and Pt layer. Therefore, very few attempts [15] were carried out to study the capped state of the bottom electrode. In the following, we will focus on analyzing the Pt morphology as well as the structure after being modified by the subsequent growth.

The subsequent growth of the capping layer on the top of the Pt films from the as-grown state involved three steps: The heating up of the samples Pt_Th10 nm, Pt_Th25 nm, Pt_Th40 nm, Pt_Th55 nm, and Pt_Th70 nm from room temperature to the growth temperature *Tg* = 900 °C with a heating rate of 25 °C/min. Then, this was followed by a growth step of the subsequent layer for a duration of 5 h, which can simultaneously be considered as the annealing step for the Pt films. The final step was a slow cooling of the bilayer systems, where the Pt films became encapsulated between the capping layer and the YSZ substrate. In the following, we compare the morphology and the microstructure only of the Pt films in the as-grown states and in the encapsulated states which were characterized at room temperature. SE_0°, EDX_0°, and BSE_0° images in Figure 6a,c,d,f compare the film morphology of the Pt films with a thickness *Th_Pt_* = 25 nm in the as-grown states, where no hillocks were detected, and in the encapsulated states, respectively. Dewetting phenomena were detected, and the size of voids increased during the post-growth stage. EDX_0° maps (Figure 6b,e) illustrate the Pt-free regions as black regions, which were significantly propagated, and the Pt film with a channel-like morphology which was transformed to a Pt block morphology. The degree of coverage was evaluated from the BSE_0° images in Appendix A, showing a decrease from 76.3% to 38.1%. Similar dewetting phenomena were measured by SEM for Pt films with a 24 nm thickness and grown by PLD at *Tg* = 780 °C on YSZ(111) after thermal annealing at 800 ° for 48 h under air and without capping layer [44].

In order to understand the dewetting mechanism occurring for the sample Pt_Th25 nm after the subsequent growth of the capping layer, corresponding TEM cross-section specimens were investigated (see Figure 6g,h). As a first result, the film thickness that corresponds to the vertical size of the formed Pt blocks was found to be around 100 nm. This indicates that a migration of the Pt nucleus took place due to high surface diffusion at *Tg* = 900 °C in addition to the desorption of Pt from the surface because of the instability of Pt when the film is still thin and does not reach the optimum thickness to withstand migration processes.

The formation of a discontinuous Pt film in the case of EncPt_Th25 nm was accompanied by an increase in void sizes from 40 nm to 150–690 nm, as illustrated by magenta arrows in Figure 6g,h of the as-grown sample Pt_Th25 nm and EncPt_Th25 nm for the corresponding encapsulated state (Table 1, lines 17, 38, column #2). The change in the film thickness was not only measured by TEM, but also by XRR curves which exhibited changes when the Pt films became encapsulated. For Pt_Th25 nm, the XRR curve displayed oscillations and the fitting of the curve was found for *Th_Pt_* = 25 nm, while for EncPt_Th25 nm the best fitting was obtained for *Th_Pt_* = 100 nm, confirming the thickening of the Pt film due to dewetting, as demonstrated by TEM imaging. Regarding the changes in the crystal structure, the HR-RSM data of Pt111 for Pt_Th25 nm and EncPt_Th25 nm (see Appendix A) are remarkably different, especially the fringe oscillations in the radial directions which disappear for EncPt_Th25 nm.

Figure 6i compares the XRR curves for *Th_Pt_* = 25 nm between the as-grown and the encapsulated states. Obviously, the XRR curves of the bilayer system such as EncPt_Th20 nm contain two critical angles *Q_Pt_* and *Q_CL_* that correspond to the Pt bottom electrode and the capping layers CLs, respectively. XRR of Pt_Th20 nm displayed oscillations where the periodicity gave a *Th_Pt_* of about 25 nm. The subsequent growth at *Tg* = 900 °C strongly affected the XRR curve, where the oscillations were completely erased. This is probably due to the strong dewetting of the Pt layer, as demonstrated by the TEM images of Figure 6g,h.

From the XRR curves of Figure 6i and the corresponding mass density profiles of Figure 6j, we derived that *Th_Pt_* = 31 nm for sample Pt_Th25 nm, whereas in the case of EncPt_Th25 nm, a thickness of *Th_Pt_* = 38 nm was found. XRR also confirmed dewetting phenomena upon encapsulation processes. Additionally, the mass density profiles of Pt_Th25 nm and EncPt_Th25 nm showed differences in the mass density distribution of the Pt layers as well as in the film thicknesses.

As indicated in Figure 6j, $\theta_{CL}$ and $\theta_{Pt}$ show the critical angles for the capping layer and the platinum layer. The comparison between the samples Pt_Th25 nm and EncPt_Th25 nm shows that the critical angle for the Pt layer is different when platinum is encapsulated. In other words, the encapsulation of platinum through the high temperature post-growth decreases the density of the Pt film (resp. degree of coverage).

Additionally, the mass density profile $\rho_{Pt}$ of the Pt_Th25 nm film did not remain constant and it varied across the film thickness up to the film surface without reaching the theoretical bulk value (indicated via the dashed line $\rho_{Pt}=21.45\;g/cm^{3},\rho_{YSZ}=5.85\;g/cm^{3}$). This mass density fluctuation correlates with the degree of coverage, where $\rho_{Pt}$ falls below the theoretical mass density of a completely covered film surface. The thickness values obtained from the density profiles are also shown in Table 1 (row#7). It should be noted that the *Th_Pt_* values obtained from the mass density profiles are inconsistent with the corresponding TEM values because of the local aspect of the TEM examination method (Table 1, rows #7, 8). Furthermore, by comparing the diffraction patterns, we found that in the case of EncPt_Th25 nm, the peak positions of the Pt111, Pt222, and Pt333 reflections shifted towards higher scattering wave vectors *Q_rad_* (see Figure 6k), which reflects a compression in the out-of-plane interplanar d_111_ spacing, i.e., reducing it from *d*_111_ = 2.9743 Å to d_111_ = 2.9713 Å.

The growth of the Pt films with different deposition times (i.e., number of shots) led to the formation of films with different morphologies. In fact, the growth mechanisms changed from a 3D island to 2D layer-by-layer growth when *Th_Pt_* exceeded 10 nm. The formation of hillocks also occurred for Pt_Th40 nm and Pt_Th55 nm to relax the stress generated in the films during growth. However, these films still contained the highest values of out-of-plane residual strain of ε_⟂_ = 0.005 (Figure 4d).

The Pt films obtained for the thicknesses *Th_Pt_* = 40 nm and 55 nm contained hillocks at room temperature, which may have formed during the cooling phase because of the difference in the thermal coefficient between the Pt film and the YSZ substrate. Figure 7 specifically illustrates the changes recorded after the post-growth of the subsequent layer in the film morphology as well as in the microstructure for the Pt film thicknesses of *Th_Pt_* = 40 nm and 55 nm, where hillocks were formed. For this purpose, we compared the SE_0°, EDX_0°, and BSE_0° images (see Figure 7a–c) of the sample Pt_Th40 nm in the as-grown state and of EncPt_Th40 nm in the encapsulated state (see Figure 7d–f). Firstly, the degree of coverage was found to increase from 80.4% to 95.3% (see Appendix A) due to the post-overgrowth of the subsequent layer, where simultaneously, BSE_0° images show that the hillock density decreased from 2.08/µm^2^ to 1.64/µm^2^ (Table 1). From BSE_0° images of Pt_Th40 nm and EncPt_Th40 nm given in Figure 7c,f**,** the mean size of the hillock width, the distance between hillocks, the channel width, as well as the hole diameters, were derived and are summarized in Table 1.

The hillocks were formed in the vicinity of the black channels which correspond to Pt-free regions. The holding of Pt_Th40 nm at *Tg* = 900 °C during the subsequent growth phase with a duration of five hours contributed to an increase in the diffusion energy of the Pt nuclei which in turn led to a reorganization of the film morphology. In our opinion, the dewetting of the Pt film in the sense of desorption of Pt atoms (i.e., detachment from the film surface) and the migration of Pt from hillocks to the film surface are two competitive processes, which occurred in the sample Pt_Th40 nm holding at *Tg* = 900 °C during the subsequent layer growth. The migration of Pt atoms belonging to overgrowth regions in the shape of bumps on the surface towards the channels led to an increase in the degree of coverage and a decrease in the hillock density. Simultaneously, the Pt-free channels, which are not surrounded by hillocks, were also attacked by dewetting phenomena which contributed to a further enlargement of the channel width during the high-temperature holding during the subsequent growth. As a result, the channel width increased from about 121 nm for Pt_Th40 nm to 237 nm for EncPt_Th40 nm, while the size of the holes changed from approximately 68 nm to 96 nm.

The increase in the holes formed was the consequence of the migration of Pt atoms toward the hillocks, which might have induced the increase in hillock size from about 179 nm to 376 nm because of local stress accumulation. This allows us to conclude that the hole expansion drives the Pt atoms to pile up around and to form hillocks near the holes with larger sizes. Similar phenomena were detected and described for the in-situ investigation of thermal instabilities and solid state dewetting in polycrystalline Pt thin films via confocal laser microscopy during the annealing of films with a free surface by Jahangir et al. [11].

Figure 7g,h show the modification of the Pt films in a cross-section by TEM micrographs for the samples Pt_Th40 nm and EncPt_Th40 nm. At first, the film thickness was found to be about 40 nm in the case of Pt_Th40 nm, while *Th_Pt_* decreased down to about 20 nm underneath the subsequent overgrown layer in the absence of hillocks. At other locations of the films, hillocks are indicated by magenta arrows with lateral sizes exceeding 200 nm and a hillock height of about 100 nm. This explains that the extension of the lateral size of the hillocks was the result of the thinning of the film thickness towards the hillocks caused by Pt migration. These results suggest that the mechanism of hillock formation is rather Pt regrowth on the top of the film surface due to the piling up of Pt atoms with high surface energy. Furthermore, the major driving forces for the Pt-film rearrangement during the subsequent growth at *Tg* = 900 °C and during the slow cooling to room temperature were rather the surface energy and the internal defect energy caused by the grain boundaries. This was also proved by Hren et al. when they discussed the hillock formation mechanisms for different types of metals such as Al, Pt, and Pb [13].

We compared the SE_0°, EDX_0°, and BSE_0° images of the sample Pt_Th55 nm in the as-grown state (Figure 7i–k) and of EncPt_Th55 nm in the encapsulated state (see Figure 7l–n). The evaluation of the BSE_0° images in Appendix A indicates an increase in the degree of coverage from 81.4% to 94.3%. The hillocks completely disappeared in EncPt_Th55 nm due to the migration of Pt nuclei into the neighboring Pt-free channels where their number decreased (see Figure 7k,n). This permits us to conclude that the competition between dewetting and hillock expansion was affected by the Pt film thickness for the same annealing duration held at *Tg* = 900 °C. As a result, the in-plane residual strain decreased in *Th_Pt_* from 40 to 55 nm (Figure 4d and Figure 5d). The film thicknesses depicted from the TEM images of Figure 7o,p do not reveal any modifications in terms of the thinning of the Pt film. This confirms that the disappearance of overgrown Pt forming the hillocks, which migrated to fill the Pt-free channels located in the neighboring regions, leading to the formation of more continuous films with some holes. The peak position of the Pt333 reflection behaved in the reverse way for the samples Pt_Th40 nm and Pt_Th55 nm as the Pt films became encapsulated.

The fitting of XRR curves was obtained for Pt_Th40 nm and EncPt_Th40 nm for *Th_Pt_* = 40 nm and 20 nm, respectively, as this could be demonstrated by the enhancement of oscillations in Figure 7q for EncPt_Th40 nm. Figure 7q compares the XRR curves for *Th_Pt_* = 40 and 55 nm in the as-grown and capsulated states. Obviously, the XRR curves of the bilayer systems such as EncPt_Th40 nm and EncPt_Th55 nm contain two critical angles, *Θ_Pt_* and *Θ_CL_*, which correspond to the Pt bottom electrode and to the capping layers (CLs), respectively. Furthermore, the oscillations recorded in the incident range Q > 0.5 degrees belong to the Pt underlayer, while the oscillations for *Θ* < 0.5 degrees correspond to the capping layer with a thickness *Th_CL_* of about 100 nm. Therefore, the simulation of XRR curves is based on one single layer for the samples Pt_Th40 and Pt_Th55 nm, but for two layers for EncPt_Th40 nm and EncPt_Th55 nm as described in the Experimental Section. All the fitting curves derived from simulations are drawn in solid lines and fitting parameters are summarized in Table 1. The increase in the oscillation period for EncPt_Th40 nm in comparison with Pt_Th40 nm was due to the thinning of the Pt layer during the subsequent growth. The Pt film thickness determined from the XRR fitting is about *Th_Pt_* = 28.18 ± 0.34 nm for EncPt_Th40 nm and *Th_Pt_* = 55.56 ± 0.9 nm for EncPt_Th55 nm (Table 1). These values were found in accordance with the *Th_Pt_* measured from the TEM images (Figure 7g,h).

To fit the XRR curves, the Pt layer was assumed to be composed of *N* slabs or slices where each slab *Nj* is defined with a thickness *ThPt_j* and mass density *ρj*. The best-fitting curves of the measured XRR curves are plotted as bold solid lines in Figure 7q. These later were obtained for the mass density profiles across the film thickness and are presented in Figure 7r. The Pt mass density *ρ_Pt_* does not display a plateau-like behavior for Pt_Th40 due to the decay of the *ρ_Pt_* across the Pt film thickness when *Th_Pt_* > 15 nm. The cut-off of the *ρ_Pt_* profile gives *Th_Pt_* = 38 nm, as is indicated by a magenta vertical dashed line in Figure 7r. The *ρ_Pt_* profile of Pt_Th40 has a maximum which is very close to the Pt mass density of the bulk (i.e., *ρ_BLPt_* = 21.45 g/cm^3^). However, the *ρ_Pt_* profile of EncPt_Th40 nm partially displays a plateau-like behavior for 15 nm < *Th_Pt_* < 38 nm. The cut-off, which is illustrated by the red vertical dashed line in Figure 7r, gives a value of about *Th_Pt_* = 42 nm where the capping layer starts. Even in the plateau-like region, the *ρ_Pt_* fluctuates and deviates from *ρ_BLPt_* = 21.45 g/cm^3^. The *ρ_Pt_* profile of Pt_Th55 nm shows a smoother curve and plateau-like behavior for 15 nm < *Th_Pt_* < 38 nm in comparison with Pt_Th40, reflecting less fluctuations in *ρ_Pt_* across the Pt layer. The mass density of the plateau is found to be *ρ_Pt_* = 18.66 g/cm^3^ which is lower than the mass density of the bulk platinum (i.e., *ρ_BLPt_* = 21.45 g/cm^3^). The cut-off, which is illustrated by the blue vertical dashed line in Figure 7r, yields a value of about *Th_Pt_* = 50 nm. For sample EncPt_Th55 nm, the *ρ_Pt_* profile has a more extended plateau-like behavior where the values fluctuate and deviate from the *ρ_BLPt_* value. The cut-off illustrated by the black vertical dashed line in Figure 7r gives a value of about *Th_Pt_* = 68 nm. It should be noted that the interfacial region between the Pt and CLs is comprises the thicknesses *Th_Pt_ =* 58 and 68 nm. The *Th_Pt_* = 65 nm determined from TEM (cf. Figure 7o) corresponds to some locations, while *Th_Pt_* = 50 nm is derived from the blue cut-off in Figure 7r and represents the average value of the Pt film thickness. This could explain the origin of the discrepancy between the determined *Th_Pt_* values. Figure 7s depicts the radial diffraction profiles of the Pt111, Pt222, and Pt333 reflections recorded from the samples in the as-grown and encapsulated states. For the higher Pt333 reflection order, the *Qz* (light red solid line) of Pt_Th40 nm is shifted toward lower *Qz* values (respectively higher *c* lattice parameter) in the case of EncPt_Th40 nm. Oppositely, the *Qz* coordinates of the Pt333 peak move to higher values when the Pt film changes from the as-grown state Pt_Th55 nm to the encapsulated state EncPt_Th55 nm. This reflects a compression of the interplanar distance *d*_111_ along the [111] direction in the rhombohedral structure space.

Figure 8a–f compare the SE_0°, EDX_0°, and BSE_0° images of Pt_Th70 nm and EncPt_Th70 nm. The Pt surface morphology, which was smooth and did not contain hillocks for Pt_Th70 nm, changed for EncPt_Th70 nm where the film morphology exhibited a continuous film with some grains formed from a second nucleation growth. In order to better understand the origin of the surface morphology modification, corresponding cross-section TEM images of the films are compared in Figure 8g,h. Here, for the sample Pt_Th70 nm, we can see that a continuous Pt film free of hillocks changed in the film thickness, leading to thickness fluctuations at specific locations of the film. Having a closer look into the subsequent layer, one can detect a few hillocks in the capping layer, which were formed on the Pt film as indicated by the red and green arrows. This confirms that the grains rather belong to the subsequent layer and not to the Pt underlayer. However, we can conclude that the film thickness of platinum also affects the growth as well as the microstructure of the subsequent layer. The fitting of XRR curves in Figure 8i revealed a mean value of *Th_Pt_* = 70 nm for Pt_Th70 nm and EnPt_Th70 nm, which confirms that *Th_Pt_* was optimum for the underlayer to withstand the dewetting phenomena during the subsequent growth at *Tg* = 900 °C.

The fitting of XRR curves is shown in Figure 8i, while Figure 8j shows the bulk-density profiles of the film. It is noticeable that the as-grown sample has a constant density profile throughout the thickness of the film, accompanied by a finely finished surface. However, it is important to note that, while the surface is sharp, the interface is not as sharply defined. Furthermore, the surface roughness of *Ra* = 0.6 nm for Pt_Th70 nm could only be derived from AFM images due to the inconvenience of the XRR several-stack model for the roughness estimation (Table 1, row #10).

In addition, the disappearance of hillocks, which is related to the increased fill factor, indirectly contributed to the sharpness of the Pt film surface. In Figure 8j, the mass density profiles of Pt_Th70 nm and EncPt_Th70 nm are consistent and the plateau of the profiles, corresponding to the as-grown state (i.e., Pt_Th70 nm) which is aligned with the theoretical density value. This is in accordance with the local filling factor obtained through BSE image analysis. It can therefore be concluded that a film thickness of 70 nm is sufficient to completely cover the YSZ(111) surface using PLD growth. On the other hand, the results revealed a slight thickness increase from *Th_Pt_* = 70 nm to *Th_Pt_* = 88 nm, which was accompanied by a small density reduction. This indicates the occurrence of a migration of Pt atoms, which did not disturb the uniformity and the continuity of the Pt film being free of holes since the mass density profile remained constant through the thickness.

Figure 8k compares the diffraction patterns of both states and demonstrates a compression in the lattice unit cell after Pt encapsulation as the *Qz* moved toward higher values in the case of EncPt_Th70 nm. The change in the microstructure of the Pt layers in the as-grown and encapsulated states was studied in terms of mosaicity, meaning the degree of misorientation and the mean value of vertical deviation <*ε_^_*>. For this purpose, the angular and radial broadening, which were derived from the HR-RSM data of Appendix A, were fitted and the corresponding *FWHM_Ang_* and *FWHM_Rad_* values are plotted in Appendix A. The upper row of Appendix A (respectively lower row) compares the WH plots for *FWHM_Ang_* (resp. *FWHM_Rad_*) as a function of the reflection order between the as-grown and encapsulated state for the Pt films.

Prior to the discussion of the change in mosaicity, as well as in the crystal lattice parameters after the post-growth of the subsequent layer, it is worthwhile to summarize the different competitive mechanisms, which occurred as the Pt film thickness *Th_Pt_* increased. Firstly, we demonstrate a morphology transition from 3D islands to 2D layer-by-layer growth when *Th_Pt_* > 10 nm. In addition, the hillock formation took place for *Th_Pt_* = 40 and 55 nm (marked in Figure 9 as “H”) when the stored stress in the grown film achieved a critical value for the appearance of the hillocks, which represents a stress relaxation mechanism occurring during the cooling stage to a certain value as measured by HR-XRD at room temperature. Combined BSE_0° and TEM imaging reliably demonstrated the influence of the diffusion energy increase on the platinum film morphology as the film was held at *Tg* = 900 °C.

In fact, dewetting of the Pt layer and migration of Pt from overgrown bumps which constituted the hillocks simultaneously occurred (see Figure 7). Therefore, for a better understanding, it is crucial to consider the Pt surface morphology, the dewetting, as well as the hillock formation, which vary from *Th_Pt_* (i.e., 10, 25, 40, 55, and 70 nm) (see Figure 9a–d).

In Figure 9, we inserted the labels “H” and “NH” (for “Hillocks” and “No Hillocks) to consider the modification of the hillocks as a function of *Th_Pt_* between the as-grown and encapsulated states. In the following, we suggest discussing the variation of the misorientation degree *α*, the mean value of the vertical strain, the distortion angle *δ*, and the in-plane residual strain *ε_//_* for the different Pt film thicknesses *Th_Pt_* (i.e., 10, 25, 40, 55, and 70 nm).

The degree of misorientation α was derived from the slopes β of the corresponding plots of *FWHM_Ang_* as a function of the reflection order using the formula α = (c/2π) × β. Figure 9a shows the variation of α for different values *Th_Pt_* with the indication of “H” and “NH” at the data points in the graph, in addition to the film morphologies “3D” or “2D”. In the as-grown state, we demonstrate that, independently from the film morphology and hillock formation, the degree of coverage evaluated from BSE_0° images of Appendix A increased and the misorientation degree α decreased due to the expansion of the Pt lateral size L_H_ (see Figure 5c) with the deposition time.

For *Th_Pt_* = 55 nm, by comparing the BSE images of Figure 7k,n corresponding to the as-grown and encapsulated states, we deduce that the hillocks disappeared, which reduced the constraint on the mosaic blocks and decreased the degree of misorientation α. A similar effect was revealed for *Th_Pt_* = 10 nm where BSE images are not shown (*Th_Pt_* = 10 nm “NH” → “H” in EncPt_Th10 nm and EncPt_Th55 nm, see Figure 9a).

However, for *Th_Pt_* = 25 nm the dewetting was found to be the dominant process during the subsequent phase and the change from Pt_Th25 nm to EncPt_Th25 nm was accompanied by a loss of coverage degree and an increase in the voids separating the Pt blocks, which in turn reduces the mosaic boundaries. For the different *Th_Pt_*, the growth of the subsequent layer represented an annealing process of 5 h duration with an annealing temperature *Tg* = 900 °C in an oxygen environment where the Pt underlayer was in an encapsulated state. This annealing contributed to a release of the mean value of the vertical strain distribution *<ε_^_>* (Figure 9b), the degree of distortion *δ* (Figure 9c), and the reduction in d-spacing *d_111_* (Figure 9d) in comparison to the as-grown state.

Our investigation proves that the encapsulation of the Pt bottom electrode with a capping layer (CL) of *Th_CL_* = 100 nm thickness reduced the hillock density for Pt_Th40 nm from 2.08 to 1.6 µm^−2^ and suppressed the hillocks for the Pt_Th55 nm sample (rows #23 and #42). This indicates that the suppression effect of the capping layer is also influenced by the Pt film morphology and by the related hillocks distribution at the interface Pt/CL, which is dependent on the Pt film thickness. Hence the grain boundaries originally involved in the hillock formation could have been annihilated or moved during the subsequent growth of 5 h at *Tg* = 900 °C.

## 4. Conclusions

Pt films were grown by PLD on YSZ111 at *Tg* = 900 °C. We investigated the influence of Pt film thickness *Th_Pt_* on the morphology as well as the Pt crystal structure and hillock formation in the as-grown state. Our investigation demonstrates the occurrence of a morphology transition from a 3D-island to 2D-layer growth as the *Th_Pt_* increases from 10 to 25 nm. By combining the results of different microscopic (i.e., SE_50 °, EDX_0°, BSE_0°, and TEM) and X-ray methods (i.e., HR-RSMs, XRR), we conclude the role of Pt thickness in the improvement of Pt film continuity, crystal quality, and the reduction in hillocks. The detailed comparison on the crystal structure and morphology of the Pt films as the function *Th_Pt_* in the as-grown and encapsulated states reveals several competitive phenomena such as hillock formation, dewetting, and coalescence, which were driven by the increase in diffusion energy at *Tg* = 900 °C and the grain boundaries and junctions. We found that *Th_Pt_* = 70 nm was an optimum thickness to obtain a good quality Pt bottom electrode, which is hillocks free and has a minor thickness fluctuation for operation at RT and at high temperatures around *Tg* = 900 °C. Our study demonstrates that even though Pt growth was performed at a high temperature *Tg* = 900 °C, hillock formation could take place dependently of the film thickness and of the resulting morphology. It is crucial to consider the Pt film rearrangement occurring during the subsequent growth, which represents an annealing process for the Pt bottom electrode. The lattice constants of the Pt films and the residual strains decreased after the subsequent growth. This indicates a compression in the unit cell and a stress release due the Pt film arrangement, which was driven by dewetting and the increase in Pt diffusion energy.

## Figures and Tables

**Figure 1 nanomaterials-14-00725-f001:**
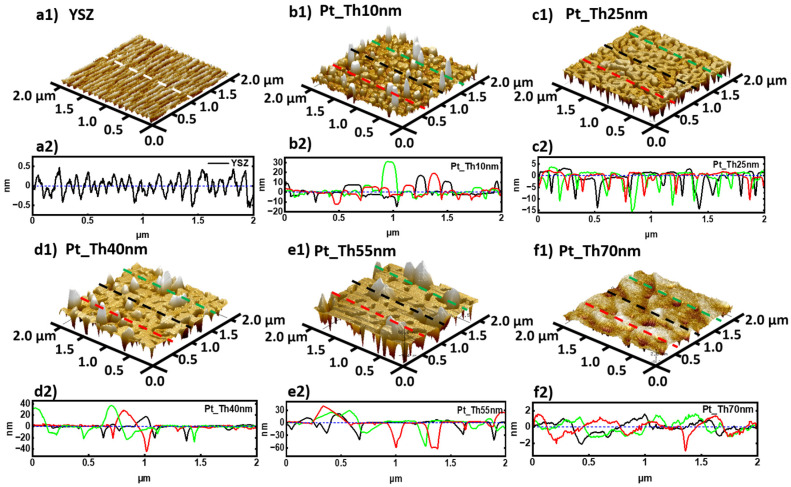
(**a1**–**f1**) and (**a2**–**f2**) are the AFM images with 2 × 2 µm^2^ sizes and the corresponding line profiles of the local height drawn by red, black, and green dashed lines are for the samples of YSZ substrate, Pt_Th10 nm, Pt_Th25 nm, Pt_Th40 nm, Pt_Th55 nm, and Pt_Th70 nm.

**Figure 2 nanomaterials-14-00725-f002:**
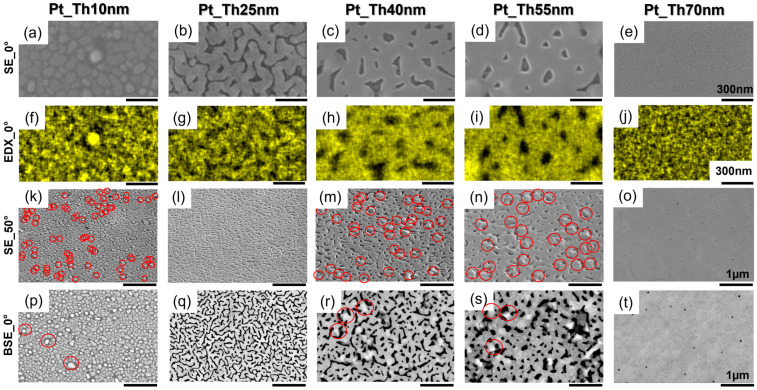
(**a**–**f**) SE_0° images recorded at zero degrees of inclination. (**f**–**j**) Corresponding EDX_0° maps and (**k**–**o**) corresponding SE_50° images taken at an inclination angle of 50 degrees. (**p**–**t**) BSE_0° images from areas with 5 × 5 µm^2^ size. All the SE_0°, EDX_0°, SE_50°, and BSE_0° data correspond to the samples Pt_Th10 nm, Pt_Th25 nm, Pt_Th40 nm, Pt_Th55 nm, and Pt_Th70 nm in the as-grown state. Red circles show the examples of the located hillocks.

**Figure 3 nanomaterials-14-00725-f003:**
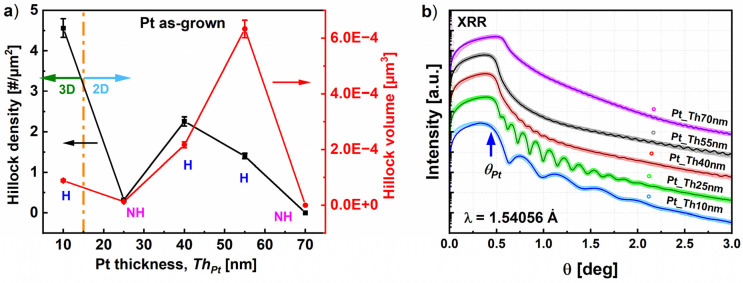
(**a**) The variation of the hillock density (left axis) and hillock volume (right axis) with the Pt thickness. The dotted line refers to the morphology transition from 3D island growth to 2D layer-by-layer growth. The labels “H” and “NH” refers to hillocks and non-hillocks, respectively, as demonstrated by AFM, SE_50°, and BSE_0° imaging. (**b**) XRR curves and the corresponding fitting for the samples Pt_Th10 nm, Pt_Th25 nm, Pt_Th40 nm, Pt_Th55 nm, and Pt_Th70 nm.

**Figure 4 nanomaterials-14-00725-f004:**
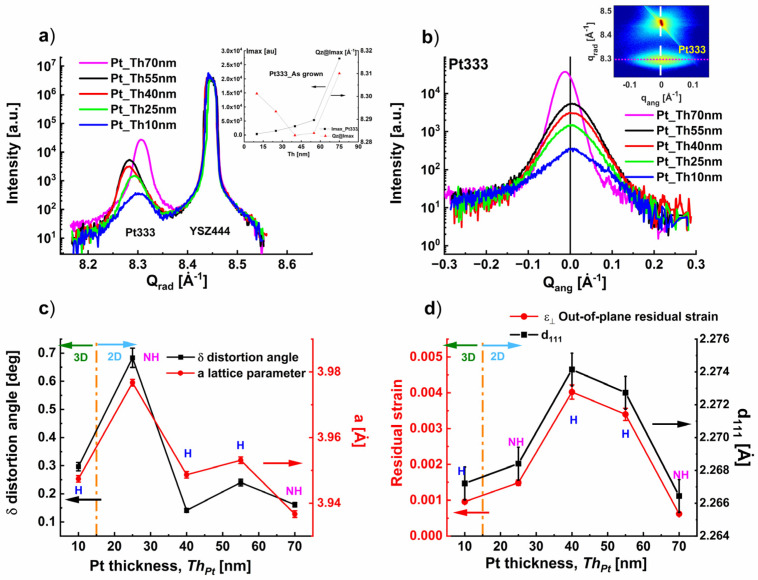
(**a**) Radial diffraction profiles of the Pt333 reflection together with the YSZ444 of the samples Pt_Th10 nm, Pt_Th25 nm, Pt_Th40 nm, Pt_Th55 nm, and Pt_Th70 nm. (**b**) Corresponding angular diffraction profiles of the Pt333 reflection. (**c**) Variation of the in-plane (left axis) and out-of-plane (right axis) lattice parameters as a function of the Pt film thickness *Th_Pt_*. (**d**) Variation of the in-plane and out-of-plane residual strain as a function of the Pt film thickness *Th_Pt_*. The labeling “H” refers to hillocks and “NH” to non-hillocks. The dotted line refers to the morphology transition from 3D island growth to 2D layer-by-layer growth when *Th_Pt_* increases up to 55 nm and then further increases from 55 to 70 nm.

**Figure 5 nanomaterials-14-00725-f005:**
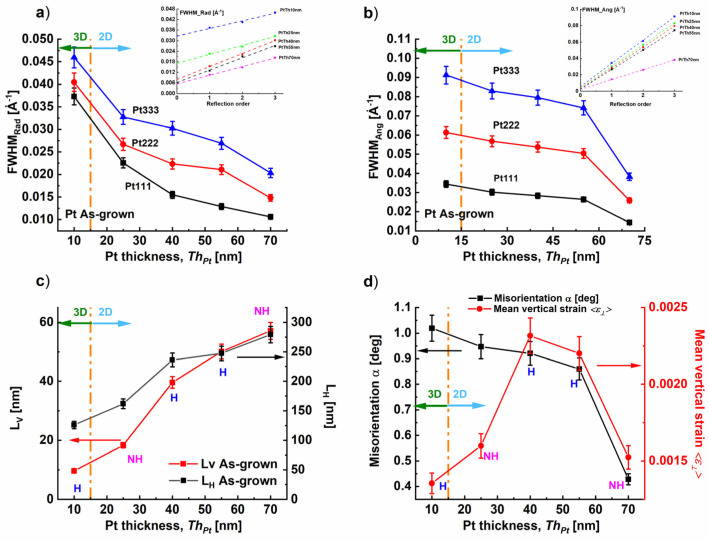
(**a**,**b**) The variation of the angular broadenings *FWHM_Rad_* and *FWHM_Ang_*, respectively, with Pt film thickness *Th_Pt_*. The corresponding insets show *FWHM_Rad_* and *FWHM_Ang_* as functions of the reflection order for the samples Pt_Th10 nm, Pt_Th25 nm, Pt_Th40 nm, Pt_Th55 nm, and Pt_Th70 nm. (**c**) The variation of the lateral *L_H_* and vertical coherence sizes *L_V_* as a function of the Pt film thickness *Th_Pt_*. The dotted line refers to the morphology transition from 3D island growth to 2D layer-by-layer growth. The labels “H” and “NH” indicate hillocks and non-hillocks, respectively. (**d**) The variation of the misorientation degree *α* and the mean vertical strain distribution <*ε*_⟂_>. The dotted line refers to the morphology transition from 3D island growth to 2D layer-by-layer growth.

**Figure 6 nanomaterials-14-00725-f006:**
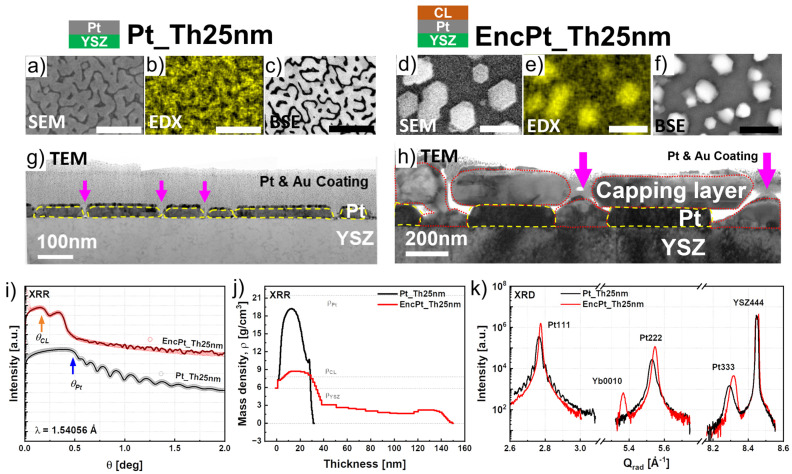
(**a**–**g**) and (**d**–**h**) SE_0°, EDX_0°, BSE_0°, and TEM images of the samples Pt_Th25 nm and EncPt_Th25 nm, respectively. In (**g**,**h**) the yellow dashed lines show the borders of the platinum film whereas the red dotted lines indicate the borders of the capping layer. Additionally, the magenta arrows indicate the holes of the platinum film. Comparison of the XRR curves of samples Pt_Th25 nm and EncPt_Th25 nm with their fitting curves (**i**), the Pt densities along the film depths (**j**), and the radial diffraction profiles (**k**) of the Pt111, Pt222, and Pt333 symmetric reflections.

**Figure 7 nanomaterials-14-00725-f007:**
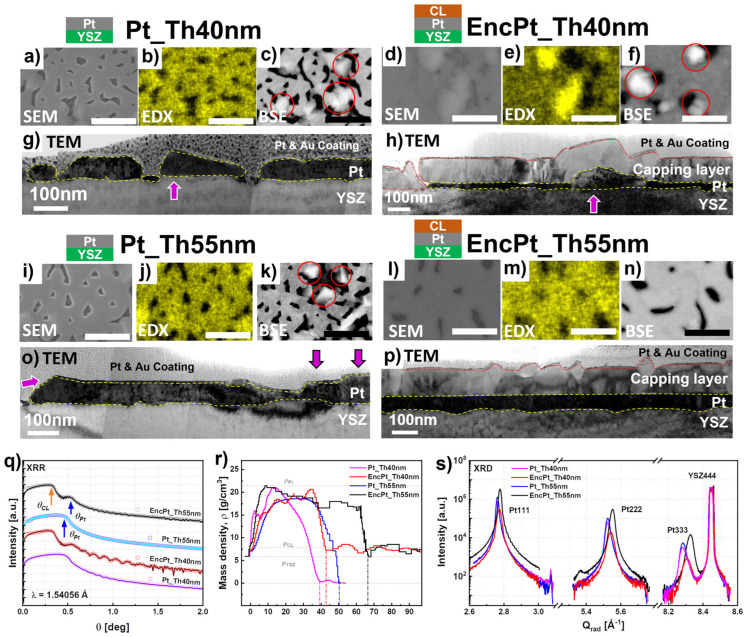
(**a**–**g**) and (**d**–**h**) SE_0°, EDX_0°, BSE_0°, and TEM images of the samples Pt_Th40 nm and EncPt_Th40 nm, respectively. (**i**–**o**) and (**l**–**p**) SE_0°, EDX_0°, BSE_0°, and TEM images of the samples Pt_Th55 nm and EncPt_Th55 nm, respectively. In (**g**,**h**,**o**,**p**) the yellow dashed lines show the borders of the platinum film whereas the red dotted lines indicate the borders of the capping layer. Additionally, the magenta arrows indicate hillock structures of the platinum film from the cross-section view. (**q**,**r**) Comparison of the XRR curves between Pt_Th40 nm and EncPt_Th40 nm and between Pt_Th55 nm and EncPt_Th55 nm with their corresponding fitting curves (**q**), Pt densities along the film depths (**r**), and the radial diffraction profiles (**s**) of the Pt111, Pt222, and Pt333 symmetric reflections.

**Figure 8 nanomaterials-14-00725-f008:**
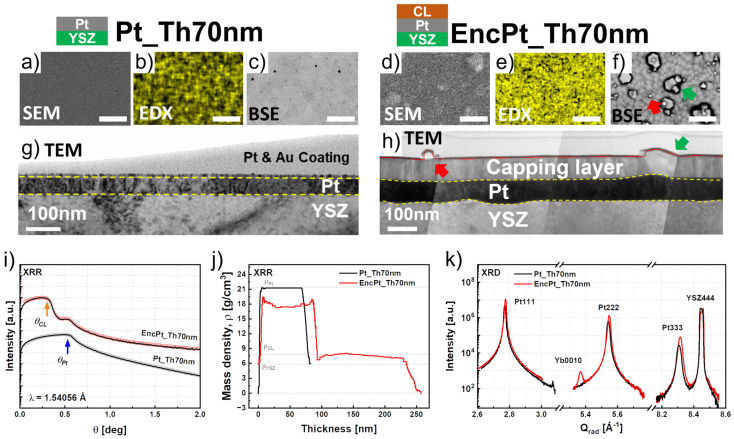
(**a**–**g**) and (**d**–**h**) SE_0°, EDX_0°, BSE_0°, and TEM images of the samples Pt_Th70 nm and EncPt_Th70 nm, respectively. In (**g**,**h**) the yellow dashed lines show the borders of the platinum film whereas the red dotted lines indicate the borders of the capping layer. Additionally, the red arrow in (**f**,**h**) indicate the second nucleation growth of the capping layer and green arrow shows the thickness fluctuation on the capping layer due to platinum hillocks. Comparison of the XRR curves of Pt_Th70 nm and EncPt_Th70 nm with their fitting curves (**i**), the Pt densities along the film depths (**j**), and the radial diffraction profiles (**k**) of the Pt111, Pt222, and Pt333 symmetric reflections.

**Figure 9 nanomaterials-14-00725-f009:**
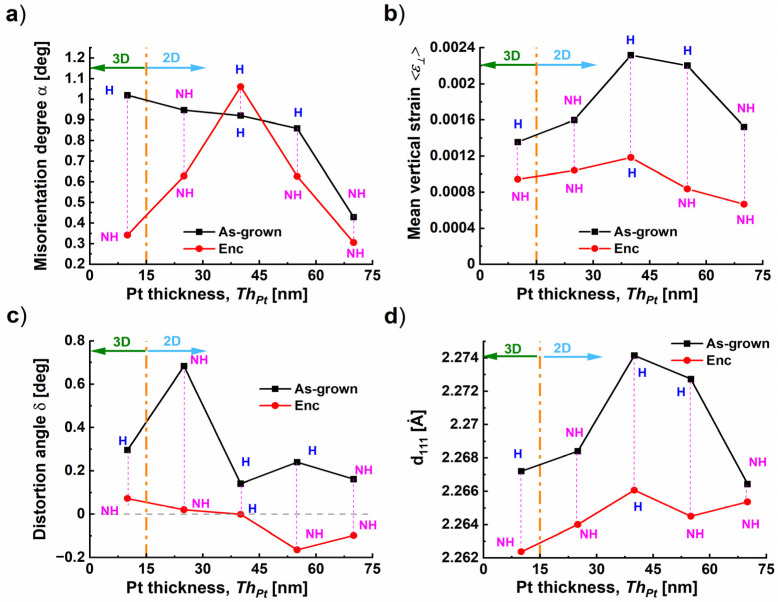
(**a**–**d**) Comparison between the as-grown and encapsulated states of Pt films with respect to the misorientation degree α (**a**), mean vertical strain distribution *<ε_^_>* (**b**), distortion angle *δ* (**c**), and d-spacing *d*_111_ along the [111] direction (**d**) as a function of the Pt film thickness *Th_Pt_*. The dotted line refers to the morphology transition from 3D island growth to 2D layer-by-layer growth. The labels “H” and “NH” refer to hillocks or non-hillocks.

**Table 1 nanomaterials-14-00725-t001:** Summary of the morphological parameters in the as-grown and encapsulated states for the Pt thicknesses *Th_Pt_* = 10, 25, 40, 55, and 70 nm.

		Column #1	Column #2	Column #3	Column #4	Column #5
**1**	**As-Grown State**	**Pt_Th10 nm**	**Pt_Th25 nm**	**Pt_Th40 nm**	**Pt_Th55 nm**	**Pt_Th70 nm**
**2**	Film thickness *Th_Pt_* [nm]	10	25	40	55	70
**3**	Topography	Island	Network	Network + Hillocks	Network + Hillocks	Network + Hole
**4**	Growth mode	3D	2D	2D	2D	2D
**5**	Mismatch [%]	−23.07	−22.25	−23.16	−22.92	−23.37
**6**	**Morphology**	**Morphology**	**Morphology**	**Morphology**	**Morphology**	**Morphology**
**7**	Thickness ^X^ [nm]	13.67	30.16	49.08	50.87	80.52
**8**	Thickness ^Y^ [nm]	NA	27.9 [23–31]	58.4 [33–72]	63.7 [30–80]	51.0 [47–55]
**9**	Roughness, Ra ^X^ [nm]	0.54 ± 0.15	1.14	1.18		0.42 ± 0.02
**10**	Roughness, Ra ^a^ [nm]	4.06	1.83	4.24	8.94	0.6
**11**	Island size ^b^ [nm]	47 [10–129]	NA	NA	NA	NA
**12**	Channel width ^a^ [nm]	NA	89 [41–141]	159 [85–247]	213 [104–286]	NA
**13**	Channel width ^b^ [nm]	NA	81 [28–142]	122 [37–308]	137 [48–376]	NA
**14**	Depth of channels ^a^ [nm]	NA	[9–25]	[36–48]	[52–65]	NA
**15**	Heights of islands ^a^ [nm]	[11–45]	NA	NA	NA	NA
**16**	**Holes**	**Holes**	**Holes**	**Holes**	**Holes**	**Holes**
**17**	Hole size ^a^ [nm]	NA	41 [15–100]	71 [30–176]	61 [18–111]	NA
**18**	Hole size ^b^ [nm]	NA	27 [7–87]	68 [24–182]	95 [28–278]	46 [24–69]
**19**	**Hillocks**	**Hillocks**	**Hillocks**	**Hillocks**	**Hillocks**	**Hillocks**
**20**	Hillocks width ^a^ [nm]	126 [88–171]	NA	187 [87–298]	276 [107–428]	NA
**21**	Hillocks width ^b^ [nm]	NA	NA	180 [117–274]	217 [99–428]	NA
**22**	Hillocks density ^a^ [/µm^2^]	4.56	NA	2.26	1.40	NA
**23**	Hillocks density ^b^ [/µm^2^]	9.44	NA	2.08	1.78	NA
**24**	Hillocks height from the film surface [nm]	21.8 [5.3–44.7]	NA	23.6 [7.2–37.0]	31.8 [16.0–52.8]	NA
**25**	Percolation					
**26**	Degree of coverage ^b^ [%]	66.2	76.3	80.4	81.4	99.8
**27**	Conductivity [×10^6^ 1/Ωm]	0	0.25	0.74	1.01	3.69
**28**	**Encapsulated state**	**EncPt_Th10 nm**	**EncPt_Th25 nm**	**EncPt_Th40 nm**	**EncPt_Th55 nm**	**EncPt_Th70 nm**
**29**	Resulting topography	Islands	Islands	Network + Hillocks	Network	Hole
**30**	Mismatch [%]	−23.68	−23.74	−23.72	−24.09	−23.93
**31**	**Morphology**	**Morphology**	**Morphology**	**Morphology**	**Morphology**	**Morphology**
**32**	Thickness ^X^ [nm]			43.72	68.6	
**33**	Thickness ^Y^ [nm]	NA	95.4 [89–109]	24.7 [19–35]	54.9 [48–73]	75.0 [70–86]
**34**	Roughness, Ra ^X^ [nm]					
**35**	Island size [nm]	NA	NA	NA	NA	NA
**36**	Channel width ^b^ [nm]	NA	NA	237 [58–1014]	226 [58–1014]	NA
**37**	**Holes**	**Holes**	**Holes**	**Holes**	**Holes**	**Holes**
**38**	Hole size ^b^ [nm]	NA	NA	97 [29–252]	95 [31–205]	NA
**39**	Hole size ^Y^ [nm]	NA	[150–690]	[45–143]	78	NA
**40**	**Hillocks**	**Hillocks**	**Hillocks**	**Hillocks**	**Hillocks**	**Hillocks**
**41**	Hillocks width ^b^ [nm]	152 [61–278]	179 [58–411]	376 [219–604]	NA	NA
**42**	Hillocks density ^b^ [/µm^2^]	NA	NA	1.64	NA	NA
**44**	Percolation					
**45**	Degree of coverage ^b^ [%]	70.5	38.1	95.3	94.3	100

a: obtained from AFM. b: obtained from BSE. s: obtained from SEM_50°. X: obtained from XRR. Y: obtained from TEM. NA: Not-Available.

## Data Availability

The data presented in this study are available on request from the corresponding author.

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
