# Peer review of "Structural and Morphological Studies of Pt in the As-Grown and Encapsulated States and Dependency on Film Thickness"

_nanomaterials, 2024, doi:10.3390/nano14080725_

Round 1
Reviewer 1 Report
Comments and Suggestions for Authors
Although Pt electrode is widely applied on the various materials, it's very important to understand the growth mechanism, so that the better Pt electrode can be prepared. The manuscript investigated the Pt growth proceed by PLD in very details. It helps us to understand this growth mechanism very well. I suggest the manuscript can be accepted.
Author Response
Dear Sir and Madame,
Please find here attached the revised version of our manuscript entitled “Structural and morphological studies of Pt in the as-grown and in the encapsulated states in dependency on the film thickness”.
For easy tracking, we include the changes in blue color considering the recommendation and the valuable proposals from the referees. Additionally, you will find a detailed replies indicating the lines number of the added explanation and references basing on the comments of the reports. We appreciate very much the careful reading of the editor and referees to the manuscript and their fruitful suggestions to improve the quality of the discussions.
Yours sincerely,
Sondes Bauer

Reviewer 2 Report
Comments and Suggestions for Authors
Authors investigated the morphology of Pt films grown by pulsed laser deposition and observed 3D-2D transition, dewetting, and hillock formation to find a proper thickness for getting a hillock-free bottom electrode, which withstands the dewetting phenomena. Comments and questions are the following.
1. Ref.[3] channel morphology was observed from Tg=700 and 900 oC. Effective thickness was 40 nm, which is thinner than 70 nm of Tg=900 oC. What are the reasons for different thickness and morphology?
2. Error bars are required in Figure 3(a), 4(c,d), 5, and S4.
3. In general authors characterized their samples systematically on crystallin structure, morphology changes, but the Pt growth using pulsed laser deposition have already been investigated by some people including Ref. [3]. They need to discuss further what are same and what are different with Pt growth on different substrates. This would be important and make their data interpretations reasonable when compared with other substrates like sapphire, Al2O3.
Author Response

(The authors gave the same response as above.)

Reviewer 3 Report
Comments and Suggestions for Authors
This manuscript shows the nanoscaled Pt film formation by PLD and the evaluation of those morphologies. Those surface structures were strongly dependent of Pt thickness, mainly observed microscopic images. The work is clear, but the scientific points why hillock was formed and it depended on Pt-thickness are unclear. I thus unfortunately could not find new scientific points in this work. The minor comments are also shown below to improve this manuscript.
1. Isn’t the surface structure depended on preparation process of Pt thin film? Different surface structures even by the same preparation way are sometimes observed. Therefore, the authors needed to confirm the reproducibility and structure differences by different preparation methodology.
2. The detailed information on apparatus is inevitable. For instance, model number, company name, and so on.
3. Why was the hillock density decreased at 20 nm Pt layer at once? Clear explanation is needed for understanding the reason.
Author Response

(The authors gave the same response as above.)
